

# Three new species of *Microlaimus* (Nematoda: Microlaimidae) from the South Atlantic

Rita C. Lima, Patrícia F. Neres and André M. Esteves

Department of Zoology, Universidade Federal de Pernambuco, Recife, PE, Brasil

## ABSTRACT

Three new species of *Microlaimus* are described from the continental shelf of the Campos Basin, southwest Atlantic, Brazil. *Microlaimus campiensis* **sp. n.** differs from all other species in the presence of two anterior testes, slender spicules with enlarged proximal ends, 7–11 pre-cloacal papilliform supplements, and females with a pair of constriction structures, one on each branch of the ovary. *Microlaimus alexandri* **sp. n.** shows sexual dimorphism in the size of the amphidial fovea, which occupies 100% of the diameter of the corresponding area in the male; the buccal cavity provided with five teeth and a slightly cuticularized cuticular ring. *Microlaimus vitorius* **sp. n.** has four longitudinal-lateral rows of glands associated with small pores, one seta and three pores small pre-cloacal, and the gubernaculum has a triangular base. An amendment to the diagnosis of the genus is proposed, where the number of teeth was modified.

## INTRODUCTION

*Microlaimus De Man, 1880* is the largest genus in the family Microlaimidae *Micoletzky (1922)*. This group is represented predominantly by marine species, with a few species occurring in brackish waters (*Tchesunov, 2014*). This genus is widely distributed across the world oceans, occurring from continental shelves (*Vanaverbeke et al., 1997*; *Muthumbi, Vanreusel & Vincx, 2011*), where it is one of the dominant genera (*Vanreusel et al., 2010*; *Moens et al., 2014*), to deep-sea regions (*Lambshead et al., 2003*; *Van Gaever et al., 2004*; *Van Gaever, 2008*); in the intertidal zone the genus is also present (*Warwick & Platt, 1973*; *Jensen, 1989*; *Kovalyev & Tchesunov, 2005*; *Leduc & Wharton, 2008*).

In a study of nematode biodiversity in sediments of the Campos Basin, South Atlantic, of which this contribution is a part, members of the genus *Microlaimus* was dominant on the continental shelf, and contributed significantly to community structuring at depths of 25–50 m on the inner shelf (*Esteves et al., 2017*; *Fonsêca-Genevois et al., 2017*).

*De Coninck & Schuurmans Stekhoven (1933)*, *Gerlach (1950)*, *Chitwood (1951)* and *Wieser (1954)* carried out the first studies on the genus and contributed lists of species and identification keys, as well as designating species that they considered dubious. Following the study by *Wieser (1954)*, several investigators have provided revisions and

Corresponding author
André M. Esteves,
andresteves.ufpe@gmail.com

information about this group and have transferred some species to other genera of the family.

*Gerlach & Riemann (1973)* organized a list of 59 species for the genus *Microlaimus*. This was reduced to 38 species following the modifications suggested by *Jensen (1978)*, which transferred 24 species of *Microlaimus* to other genera (*Aponema Jensen, 1978*; Bolbolaimus *Cobb, 1920*; Calomicrolaimus *Lorenzen, 1976*; Molgolaimus *Ditlevsen, 1921* and *Prodesmodora Micoletzky, 1923*). The genus *Calomicrolaimus*, which was comprised of 12 species, was considered a junior synonym of *Microlaimus* (*Kovalyev & Tchesunov, 2005*); however, *Tchesunov (2014)* reestablished the validity of *Calomicrolaimus*, yet only *C. rugatus Lorenzen, 1976* was considered to be a species of this genus and the other species remained in the *Microlaimus* species composition. *Leduc (2016)* updated the list of species presented in *Kovalyev & Tchesunov (2005)* study and suggested some modifications, including the transfer of two species from the genus *Aponema* to *Microlaimus*, taking the absence of dorsal apophysis into consideration, since this characteristic (presence of apophysis in the gubernaculum) is a diagnostic characteristic of *Aponema* (*Jensen, 1978*; *Tchesunov, 2014*). The structure of the female reproductive system caused *Lorenzen (1994)* and *Shi & Xu (2016)* to carry out new combinations involving the genera *Microlaimus* and *Molgolaimus*, thus species with outstretched, ovaries remain in the genus *Microlaimus*, with this being one of the diagnostic characteristics of the Microlaimidae family (*Decraemer & Smol, 2006*).

## MATERIALS AND METHODS

The Campos Basin (23°30'S and 21°30'W) extends from the states of Rio de Janeiro to Espírito Santo, with the northern margin defined by the Espírito Santo Basin, and the southern margin defined by the Santos Basin. The basin covers an area of approximately 120,000 km$^2$ and reaches depths of 3,500 m. The region is influenced by the Brazil Current, which flows parallel to the coast and reaches depths of 200 m. The basin floor is covered with fine continental sediment and sand, composed mainly of foraminiferans (*Soares-Gomes et al., 1999*).

Sampling was conducted during the cruises carried out by the HABITATS project; May 2008 and July 2009. Sediment samples were collected with a Van Veen grab (dimensions 92 × 80 × 40 cm). The sediment samples were washed through sieves with openings of 500 and 45 μm, and the material retained on the smaller-mesh sieve was passed through the flotation technique with a solution of colloidal silica (*Somerfield, Warwick & Moens, 2005*).

Nematodes were gently picked out with a stainless-steel stylet, fixed in 4% formaldehyde, and transferred to glycerin (*De Grisse, 1969*). Drawings were made using an OLYMPUS CX 31 optical microscope fitted with a camera lucida. Photographs were taken with a Zeiss AxioCam ICc 5 digital camera. The software ZEN lite 2012 was used for image processing.

The diagnosis is an amendment to that of *Decraemer & Smol (2006)*. The holotype and one paratype (female) of each species are deposited in the Nematoda Collection of Museum of Oceanography Prof. Petronio Alves Coelho (MOUFPE), Brazil. Other

**Table 1 Abbreviations for body structures used in the tables, after _Coomans (1979)_.**

| Abbreviations | Body regions |
| --- | --- |
| a, b, c | _De Man (1880)_ ratios: |
| | a—Body length divided by maximum body diameter |
| | b—Body length divided by pharynx length |
| | c—Body length divided by tail length |
| abd | Anal/cloacal body diameter |
| amph | Amphidial fovea diameter |
| amph. pos | Distance of amphidial fovea from anterior end |
| Amph% | Percentage of amphideal fovea diameter in relation to corresponding body diameter |
| blb | Pharynx bulb diameter |
| blb % | Percentage of pharynx bulb diameter in relation to corresponding body diameter |
| cbd | Corresponding body diameter |
| 2$^{nd}$ ceph ccl | Second cephalic circle |
| cs | Length of cephalic setae |
| cs/hd | Length of cephalic setae in relation to cephalic diameter |
| els | Length of external labial setae or papillae |
| exc. p | Distance of secretory-excretory pore from anterior body end |
| exc. p% | Secretory-excretory pore position in relation to pharynx length, percentage |
| gub | Gubernaculum length |
| gub/spic | Gubernaculum proportion in relation to spicule length |
| hd | Cephalic diameter |
| L | Body length |
| mbd | Maximum body diameter |
| mbd/hd | Maximum body diameter in relation to cephalic diameter |
| n. ring | Position of nerve ring from anterior body end |
| n. ring% | Nerve ring position in relation to pharynx length, percentage |
| ph | Pharynx length |
| spic | Length of spicules along arc |
| spic/abd | Spicule proportion in relation to body diameter at level of cloaca |
| t | Tail length |
| t/abd | Tail length in relation to body diameter at level of anus/cloaca |
| v | Distance of vulva from anterior end of body |
| V% | Position of vulva as percentage of body length from anterior end |

paratypes are deposited in the Meiofauna Laboratory, Zoology Department, Federal University of Pernambuco (NM LMZOO-UFPE). The nomenclature adopted for the body regions is presented in Table 1 and the measurements are expressed in micrometers.

The electronic version of this article in Portable Document Format (PDF) will represent a published work according to the International Commission on Zoological Nomenclature (ICZN), and hence the new names contained in the electronic version are effectively published under that Code from the electronic edition alone. This published work and the nomenclatural acts it contains have been registered in ZooBank, the online registration

system for the ICZN. The ZooBank LSIDs (Life Science Identifiers) can be resolved and the associated information viewed through any standard web browser by appending the LSID to the prefix http://zoobank.org/. The LSID for this publication is: urn:lsid:zoobank.org: pub: B4106DEE-2BC2-48D2-9BE5-41573F76FB25. The online version of this work is archived and available from the following digital repositories: PeerJ, PubMed Central and CLOCKSS.

The list of 87 valid species for the genus *Microlaimus* (Appendix) was based on studies by *Lorenzen (1994)*, *Kovalyev & Tchesunov (2005)*, *Leduc (2016)*, *Shi & Xu (2016)* and *Bezerra et al. (2021)*.

## RESULTS

**Systematics**
**Class Chromadorea** *Inglis, 1983*
**Subclass Chromadoria** *Pearse, 1942*
**Order Microlaimida** *Leduc, Verdon & Zhao, 2018*
**Superfamily Microlaimoidea** *Micoletzky, 1922*
**Family Microlaimidae** *Micoletzky, 1922*
**Genus** *Microlaimus De Man, 1880*
(Syn *Microlaimoides Hoeppli, 1926*; *Paracothonolaimus Schulz, 1932*)

**Diagnosis.** (emended from *Decraemer & Smol, 2006*) Cuticle transversely striated, punctuations or longitudinal bars may be present. Lateral differentiation in the form of lateral alae occurs in *M. falciferus Leduc & Wharton, 2008*. Cephalic region often set off. Epidermal glands associated or not with pores or setae, small somatic setae occur in some species. Anterior sensilla arranged according to pattern 6 + 6 + 4: six inner labial setae, usually papilliform; six external labial setae, papilliform or setiform; and four cephalic setae. Cephalic setae longer than external labial setae, except for *M. discolensis Bussau, 1993*, in which setae are equal in length. Amphidial fovea cryptocircular or unispiral (= cryptospiral), usually located near cephalic setae. Sexual dimorphism in amphidial fovea size present or absent, when present it is larger in male. Buccal cavity small to medium-sized, with three small or well-developed teeth, especially dorsal tooth, except for *M. alexandri* sp. n., where five teeth are observed (one larger dorsal tooth located in anterior portion of buccal cavity, another dorsal tooth situated in median region; one of ventrosublateral teeth located at same level as previous dorsal tooth, and others located at basis of oral cavity), making it necessary to amend the diagnosis of the genus. Transverse cuticularized band or ring may be present in buccal cavity. Most species with two testes extending in opposite directions; some with two anterior testes, others with only one testis, positioned anteriorly or posteriorly. Pre-cloacal supplements absent or present (papilliform, tubular, or small pores). Spicules usually arcuate (1–2x cloacal diameter). Gubernaculum without dorso-caudal apophysis. Female didelphic-amphidelphic, with outstretched ovaries. Tail conical, predominantly.

**Type species:** *Microlaimus globiceps De Man, 1880*.

### *Microlaimus campiensis* sp. n.
(Figs. 1–3; Tables 2 and 3)

**Type material.** Holotype male (MOUFPE 0007), paratype female (MOUFPE 0008), 3 male paratypes (430–432 NM LMZOO-UFPE) and 4 female paratypes (433–436 NM LMZOO-UFPE).

**Type locality.** Campos Basin (Rio de Janeiro, Brazil). Holotype male and female: 22°11′20″ S, 040°91′27″W (25 m depth), July 2009.

**Etymology.** The specific epithet '*campiensis*' refers to the Campos Basin, where the species was collected.

**Description. Holotype male** (Figs. 1 and 3; Table 2). Body 813 µm long. Maximum body diameter corresponding to 1.8× cephalic diameter. Cuticle striated posteriorly to cephalic setae insertion. Anterior sensilla arrangement consisting of six inner labial papillae, six external labial papillae and four cephalic setae, in different cycles. Cephalic setae corresponding to 46% of cephalic diameter or 7.5× length of external labial setae. Amphidial fovea cryptocircular, located immediately posterior to buccal cavity; occupying 28.5% of corresponding body diameter. Buccal cavity cuticularized, cheilostoma with twelve longitudinal folds. Three cuticularized teeth, one large dorsal tooth and two smaller ventrosublateral, teeth. Pharynx dilated around buccal cavity and terminal oval bulb occupying 78% of corresponding body diameter. Nerve ring located at level of secretory-excretory pore, corresponding to 60% of pharynx length from anterior end. Ventral gland located immediately posterior to pharynx. Reproductive system with two anteriorly directed testes of approximately equal size. Spicules slender and enlarged proximally, about 1.3× cloacal diameter. Gubernaculum slender, about 2.2× spicule length. Seven pre-cloacal papilliform supplements, associated glands were observed only in papillae close to the cloaca. Tail conical, about 3× cloacal diameter. Three caudal glands.

In paratype males, the external labial papillae and nerve ring were difficult to observe. The buccal cavity, cuticularization of teeth, spicules shape and pre-cloacal supplements were similar among paratype males.

**Paratypes females** (Figs. 2 and 3; Table 2). Female similar to male. Body 652–858 µm long and maximum diameter 45–51 µm. Body wider than in male. Cuticle striated from cephalic setae insertion. The external labial papillae are difficult to view; cephalic setae corresponding to 37–45% of cephalic diameter. Amphidial fovea cryptocircular, positioned similarly to male, immediately posterior to buccal cavity, 14–19 µm from anterior end and occupying about 27–31% of corresponding body diameter. Buccal cavity and teeth similar to those of male. Pharynx similar to that of male, with terminal bulb occupying 82–87% of corresponding body diameter. Nerve ring not observed. Secretory-excretory pore occupying position similar to that of male, 81–94.4 µm from anterior end and equivalent to 53–65% of pharynx length. Ventral gland located immediately posterior to pharynx. Tail with same shape and measurements as in male. Three caudal glands. Reproductive system didelphic-amphidelphic, outstretched ovaries located to right of

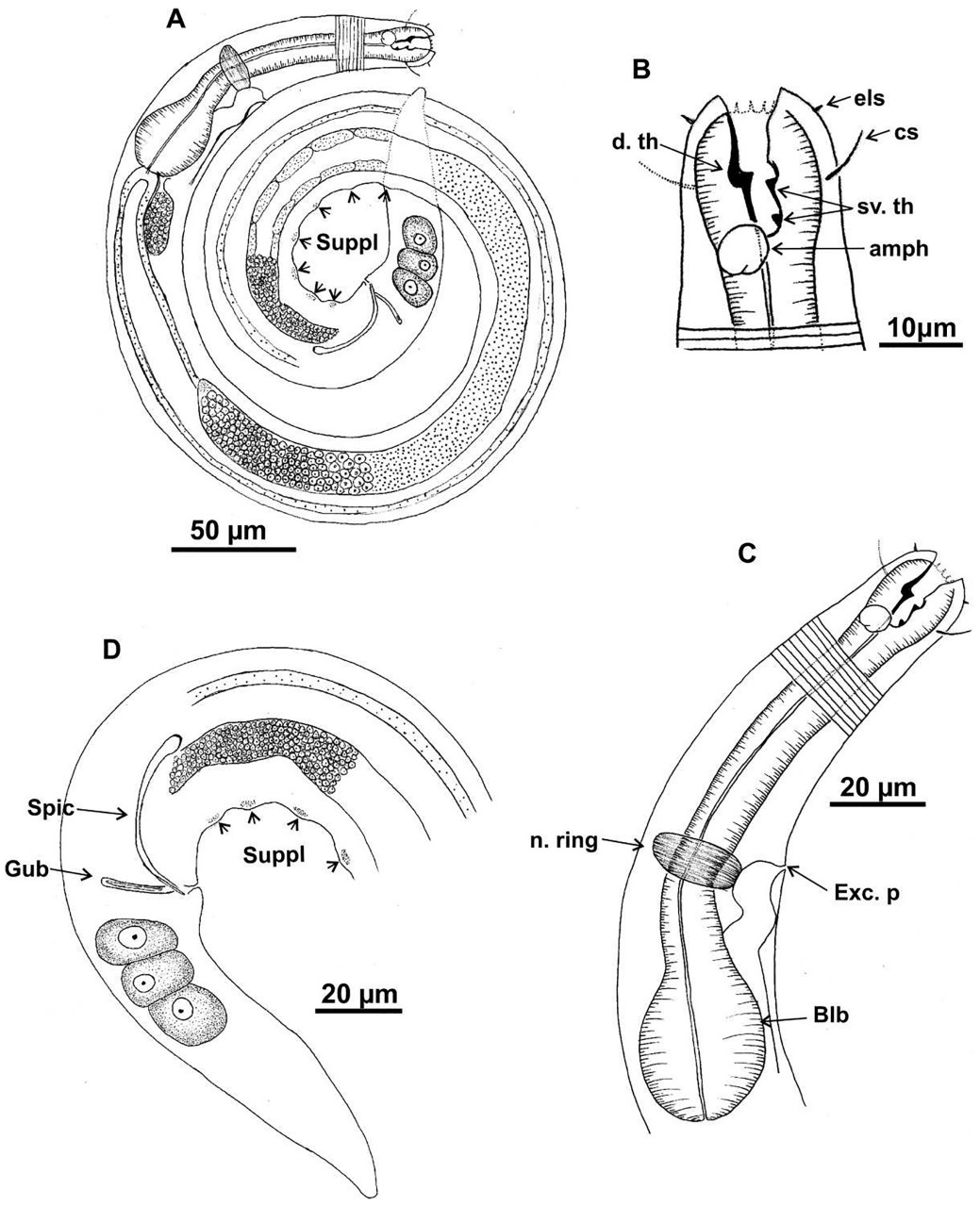

**Figure 1** *Microlaimus campiensis* **sp. n.** Holotype (male): (A) habitus; (B) anterior end (buccal cavity, amphidial fovea); (C) anterior region (cuticle, secretory-excretory pore, nerve ring and bulb); (D) posterior region (tail, spicule, gubernaculum and pre-cloacal supplements). **els:** external labial setae, **cs:** cephalic setae, d. th: dorsal tooth; sv. th: ventrosublateral teeth; amph: amphidial fovea; Exc. p: secretory-excretory pore; n. ring: nerve ring; blb: bulb; Gub: gubernaculum; Spic: spicule; Suppl: pre-cloacal supplements.

intestine. Anterior and posterior genital branches measuring 152–165 and 143–180 μm, respectively. Vulva located 348–458 μm from anterior end, 50–54% of body length. They present a structure in each ovary branch which is most easily visualized in the dorsal and ventral portions of the body (Figs. 2A–2D). In another female paratype (Fig. 3H),

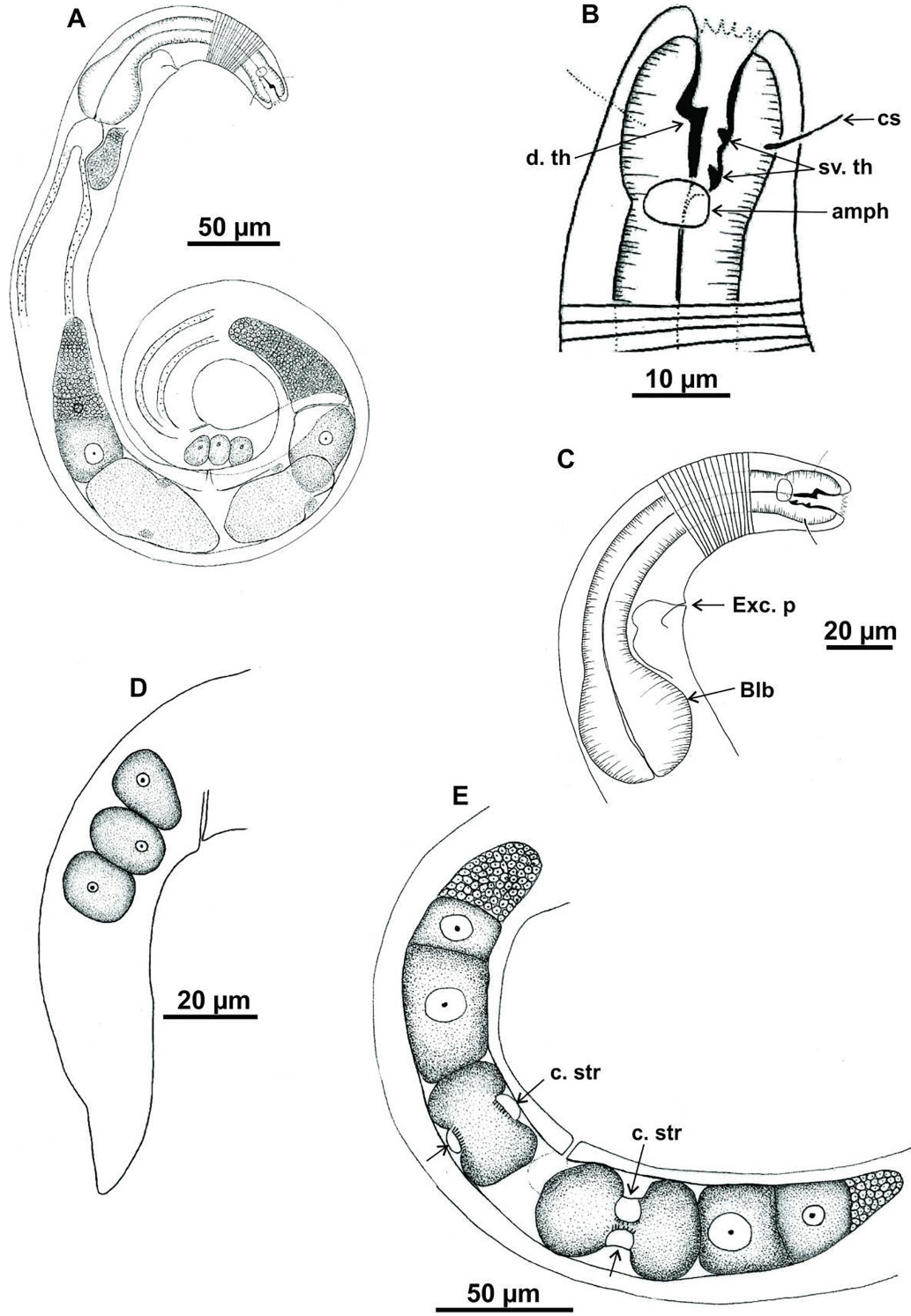

**Figure 2** *Microlaimus campiensis* **sp. n.** Paratype female (MOUPE 0008): (A) habitus; (B) anterior end (buccal cavity, amphidial fovea); (C) anterior region (cuticle, secretory-excretory pore and bulb); (D) posterior region (tail). Female paratype: (E) ovaries (contriction structures). cs: cephalic setae; d. th: dorsal tooth; sv. th: ventrosublateral teeth; amph: amphidial fovea; Exc. p: secretory-excretory pore; blb: bulb; c. str: constriction structures.               

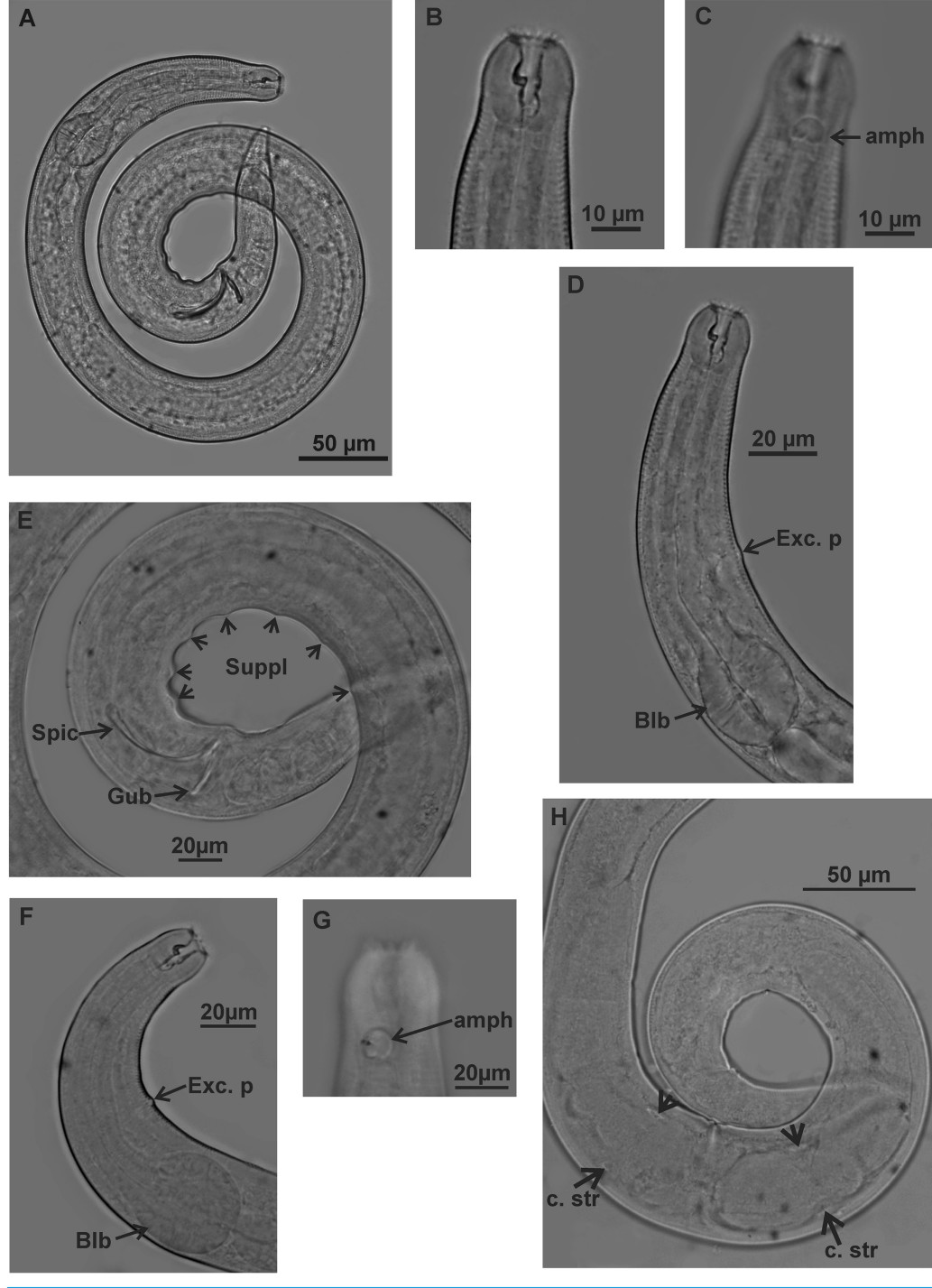

**Figure 3 *Microlaimus campiensis* sp. n.** Holotype (male): (A) habitus; (B) buccal cavity (C) amphidial fovea; (D) anterior region (secretory-excretory pore and bulb); (E) posterior region (spicules, guberna-culum and pre-cloacal supplements). Paratype female (MOUPE 0008): (F) anterior region (secretory-excretory pore and bulb); (G) amphidial fovea; (H) ovaries (constriction structures). amph: amphidial fovea; Exc. p: secretory-excretory pore; blb: bulb; Gub: gubernaculum; Spic: spicule; Suppl: pre-cloacal supplements; c. str: constriction structures.

**Table 2 Body measurements (µm) of sp. n.**

| | Holotype (Male) | Paratype Male 1 | Paratype Male 2 | Paratype Male 3 | Paratype Male 4 | Paratype Female (MOUFPE 0008) | Paratype Female 1 | Paratype Female 2 | Paratype Female 3 |
|---|---|---|---|---|---|---|---|---|---|
| L | 813 | 881 | 798 | 813 | 864 | 858 | 825 | 843 | 653 |
| mbd | 34 | 42 | 38 | 37 | 40 | 51 | 48 | 59 | 45 |
| mbd/hd | 1.8 | 2.1 | 2 | 1.9 | 1.9 | 2.4 | 2.1 | 3 | 2.4 |
| a | 23.5 | 21 | 20.8 | 21.7 | 21.5 | 16.6 | 14.5 | 14.2 | 16.7 |
| b | 5.7 | 6.1 | 6 | 5.9 | 5.8 | 5.6 | 5.6 | 5.3 | 4.7 |
| c | 8.5 | 8.9 | 8.6 | 8.5 | 9 | 8.9 | 7 | 9.4 | 8.2 |
| amph. pos | 18 | 17 | 13 | 14 | 22 | 17 | 19 | 16 | 14 |
| amph | 6 | 7 | 7 | 7 | 6 | 7 | 6 | 6 | 6 |
| cbd/amph | 21 | 19 | 18 | 25 | 20 | 21 | 22 | 19 | 20 |
| Amph% | 28.5% | 34% | 37% | 26% | 29% | 31% | 27% | 31% | 29% |
| hd | 19 | 20 | 19 | 20 | 21 | 21 | 23 | 20 | 19 |
| els | 1 | NO | NO | NO | NO | NO | 1 | NO | NO |
| cs | 9 | NO | 10 | 12 | 5 | 8 | 10 | 9 | NO |
| cs/hd | 0.5 | NO | 0.5 | 0.6 | 0.3 | 0.4 | 0.5 | 0.5 | NO |
| ph | 141 | 142 | 133 | 138 | 149 | 154 | 157 | 158 | 136 |
| blb | 28 | 27 | 26 | 35 | 35 | 36 | 41 | 37 | 29 |
| cbd/blb | 36 | 34.2 | 33 | 40.2 | 39.6 | 41.4 | 47.8 | 44.8 | 33.6 |
| blb% | 78% | 79% | 80% | 86.5% | 89% | 87% | 85% | 82% | 86 |
| n. ring | 84 | 87 | NO | NO | NO | NO | NO | NO | NO |
| n. ring% | 59.5% | 61% | NO | NO | NO | NO | NO | NO | NO |
| exc. p | 84 | NO | 79 | 83 | NO | 81 | 94 | 94 | 81 |
| exc. p% | 59.5% | NO | 59% | 60% | NO | 53% | 65% | 60% | 59.5% |
| abd | 31 | 37 | 34 | 33 | 30 | 30 | 33 | 32 | 28 |
| spic | 40 | 43 | 38 | 38 | 43 | NA | NA | NA | NA |
| gub | 18 | 20 | 19 | 15 | 18 | NA | NA | NA | NA |
| v | NA | NA | NA | NA | NA | 445 | 409 | 458 | 348 |
| V% | NA | NA | NA | NA | NA | 52% | 50% | 54% | 53% |
| t | 96 | 99 | 93 | 96 | 96 | 97 | 94 | 90 | 80 |
| t/abd | 3 | 2.6 | 2.8 | 2.9 | 3.2 | 3.2 | 2.9 | 2.8 | 2.9 |

Note:
NA = not applicable; NO = not observed. See Table 1 for abbreviations.

**Table 3 Comparisons between *Microlaimus campiensis* sp. n. and morphologically similar species (only males).**

| Species | L | a | b | c | 2nd ceph ccl | Amph% | cs/hd | hd/mbd | spic/abd |
|---|---|---|---|---|---|---|---|---|---|
| *M. campiensis* **sp. n.** | 800–880 | 20–23.5 | 5.7–6 | 8.5–8.9 | papilliform | 28.5 | 0.5 | 1.8 | 1.3 |
| *M. affinis* | 710–892 | 22.9–26 | 6.1–7.1 | 9–10.3 | papilliform | 34 | 0.4 | 1.9 | 1 |
| *M. cyatholaimoides* | 700–1,000 | 22–31 | 6.8-7.8 | 9.7–12.7 | papilliform | 35 | 0.4 | 2.9–3.6 | 1.5 |
| *M. lunatus* | 1,200–1,300 | 41 | 8.3 | 14 | setiform | 47–54 | 0.5 | 2.3–2.5 | 1.6–1.8 |
| *M. papillatus* | 960 | 44 | 7 | 11.3 | setiform | 65 | 0.5 | 1.3 | 1.3 |

Note:
For abbreviations see Table 1.

these structures were observed to cause ovary constriction, which may be related to egg expulsion.

**Diagnosis.** Anterior sensilla arrangement consisting of six inner labial papillae, six external labial papillae and four cephalic setae (6+6+4). Cephalic setae corresponding to 26–60% of cephalic diameter. Amphidial fovea, located immediately posterior to buccal cavity, accounting for 26–37% of the corresponding body diameter. Buccal cavity with three teeth, one large dorsal and two smaller ventrosublateral. Males are characterized by two testes anteriorly positioned, slender spicules with cephalized proximal region (1.1–1.4× cloacal diameter) and 7–11 papilliform pre-cloacal supplements, of which some appear to be connected to the glands. Females with a pair of constriction structures, one on each branch of the ovary.

**Differential diagnosis** (Table 3). *Microlaimus campiensis* **sp. n.** resembles *M. affinis Gerlach, 1958* in the buccal cavity, both conspicuously large, heavily cuticularized and with large teeth; the body length (798–880 *vs* 710–892 in *M. affinis*) and external labial circle (papilliform). However, *M. campiensis* **sp. n.** can be differentiated from *M. affinis* by the cephalic setae length (46–50% of cephalic diameter *vs* 37–43% in *M. affinis*). *Microlaimus campiensis* **sp. n.** has thin spicules, measuring between 40–43 μm (1.3× cloacal diameter), with a cephalized proximal region, while in *M. affinis* the spicules are lamellar, not cephalized in the proximal region, and its length is 24 μm (1.0× cloacal diameter). The new species has supplements (not described for *M. affinis*). There are also some variations in the tail length (93–96 μm in *M. campiensis* **sp. n.**; 77–87 μm in *M. affinis*) and "c" ratio (8.5–8.9 in *M. campiensis* **sp. n.**; *M. affinis* 9.0–10.3).

*Microlaimus campiensis* **sp. n.** is similar to *M. lunatus* (*Wieser & Hopper, 1967*) in the buccal cavity with well-developed teeth, position of the amphidial fovea (amph. pos/hd <1.5), and papilliform supplements. However, the new species possesses distinct characteristics from those of *M. lunatus*, such as a shorter body (798–880 *vs* 1,200–1,300 μm in *M. lunatus*) and greater width (De Man ratio a = 21–23 *vs* 39–41 in *M. lunatus*,), a smaller amphidial fovea (28.5% *vs* 47–54% of corresponding body diameter). In addition, *M. lunatus* has six pairs of caudal setae, a characteristic not observed in *M. campiensis* **sp. n.**, and the spicules measure 1.7–1.8× body diameter at level of cloaca, *vs* 1.3× in *M. campiensis* **sp. n.**.

*Microlaimus campiensis* **sp. n.** and *M. papillatus Gerlach, 1956* are similar in the broad, heavily cuticularized buccal cavity with well-developed teeth; body length (798–880 *vs* 960 μm in *M. papillatus*), cephalic setae length (46% *vs* 50% of the cephalic diameter in *M. papillatus*), the position of the amphidial fovea (amph. pos/hd <1.5) and papilliform pre-cloacal supplements. However, *M. campiensis* **sp. n.** has a wider body than *M. papillatus* (De Man ratio a = 21–23 *vs* 44), in cephalic sensilla arrangement, the external labial circle is papilliform (1 μm) in the new species and setiform (5 μm) in *M. papillatus*, and the diameter of the amphidial fovea expressed as percentage of corresponding body diameter (%) (26–37% *vs* 65% in *M. papillatus*). In addition, the spicules in

*M. campiensis* **sp. n.** are longer than in *M. papillatus* (37.8–42.6 μm *vs* 26 μm in *M. papillatus*).

  *Microlaimus campiensis* **sp. n.** differs from all the above-mentioned species in having two testes anteriorly positioned, one to the right of the intestine; it was difficult to observe the position of the other testis relative to the intestine. Two anteriorly positioned testes is a feature shared only with *M. cyatholaimoides De Man, 1922*, reported in the redescription by *Pastor de Ward (1989)*. *Microlaimus campiensis* **sp. n.** is similar to *M. cyatholaimoides* in the body length (798–880.5 *vs* 700–1,000 in *M. cyatholaimoides*), and in the values of De Man ratio a (20–23.5 *vs* 22–31 in *M. cyatholaimoides*). *Microlaimus cyatholaimoides* differs from *M. campiensis* **sp. n.** in having four lateral rows of hypodermal glands, which are absent in *M. campiensis* **sp. n.;** in the position of the amphidial fovea (amph. pos/hd <1.5 in *M. campiensis* **sp. n.** *vs* ≥ 1.5 in *M. cyatholaimoides*), De Man index c (8.5–8.9 in *M. campiensis* **sp. n.** *vs* 9.7–12.7 in *M. cyatholaimoides*) and pre-cloacal supplements (papilliform in *M. campiensis* **sp. n.** *vs* pores in *M. cyatholaimoides*).

  In females of *M. campiensis* **sp. n.,** each ovary has a constriction structure that constricts the egg, suggesting a strategy for facilitating the expulsion of eggs. These constriction structure were not observed in females of other *Microlaimus* species.

### *Microlaimus alexandri* sp. n.
(Figs. 4–6; Tables 4 and 5)

**Type material.** Holotype male (MOUFPE 0009), paratype female (MOUFPE 0010), 1 male paratype (437 NM LMZOO-UFPE).

**Type locality.** Campos Basin, Rio de Janeiro, Brazil. Holotype male and paratype female: 21°18′39″S 40°47′49″W (25 m depth), July 2009.

**Etymology.** The specific epithet '*alexandri*' is given in honor of the husband of the first author, Alexandre de Aguiar Góes.

**Description. Holotype male** (Figs. 4 and 6; Table 4). Body 1,146 μm long. Maximum body diameter corresponding to 2.7× cephalic diameter. Cephalic set off. Cuticle thin, transversely striated from cephalic setae insertion. Cuticle striations more visible in anterior portion of body. Anterior sensilla arrangement consisting of six inner labial papillae, six external labial papillae and four cephalic setae, in different cycles. Cephalic setae corresponding to 57% of cephalic diameter. Amphidial fovea unispiral, located posterior to buccal cavity, and occupying 100% of corresponding body diameter. Buccal cavity cuticularized, with folds in its first portion. Five teeth, two dorsal and three ventrosublateral: larger dorsal tooth located in anterior portion of buccal cavity, another dorsal tooth situated in median region; one of ventrosublateral teeth located at same level as previous dorsal tooth, and others located at basis of oral cavity. Buccal cavity with slightly cuticularized cuticular ring, positioned at the level of larger dorsal teeth base. Pharynx with terminal oval bulb. Nerve ring located at 67% of pharynx length. Secretory-excretory pore not observed. Male reproductive system consisting of two

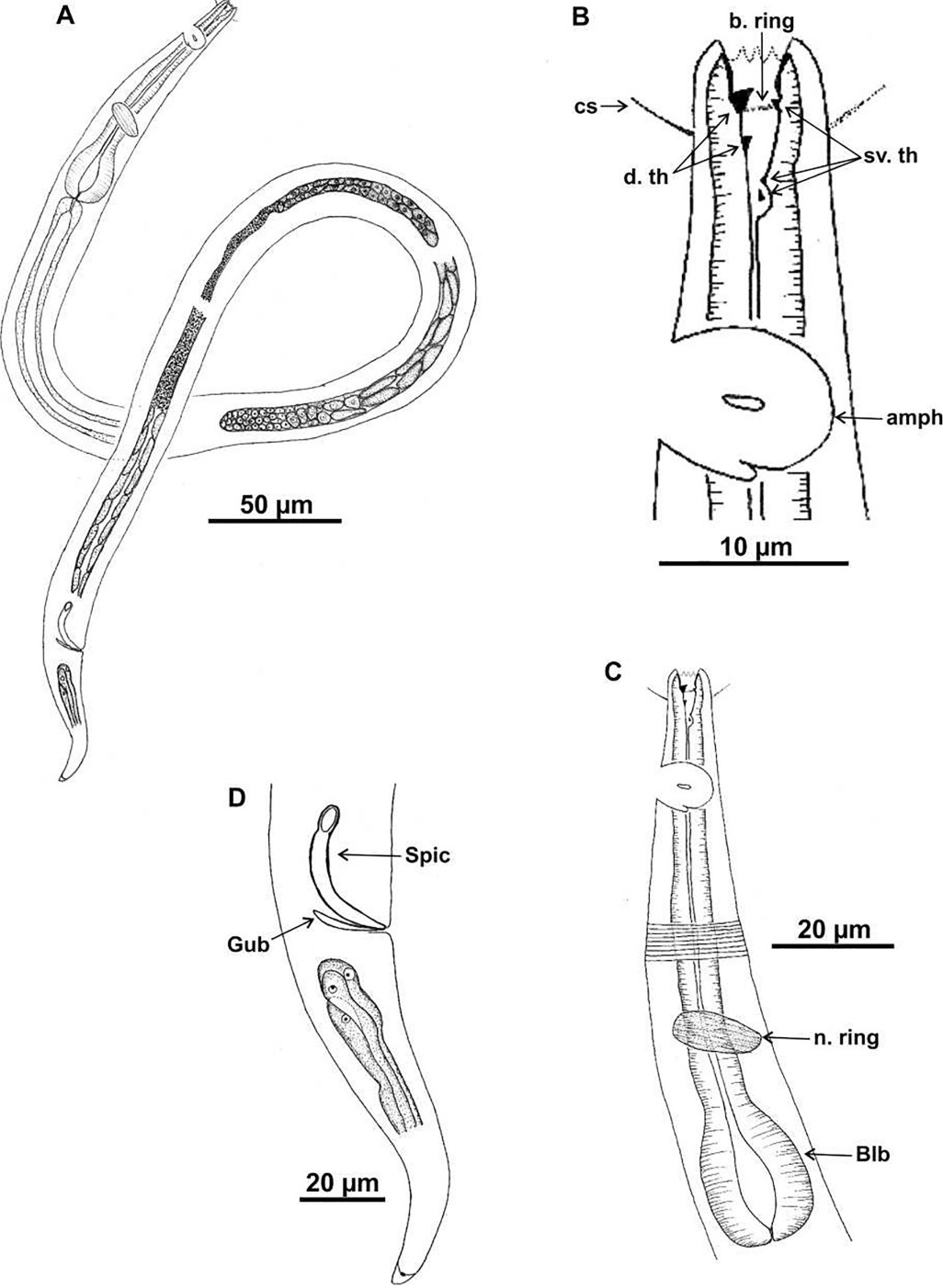

**Figure 4 *Microlaimus alexandri* sp. n.** Holotype (male): (A) habitus; (B) anterior end (buccal cavity, cephalic setae and amphidial fovea); (C) anterior region (cuticle, nerve ring and bulb); (D) posterior region (tail, spicule and gubernaculum). cs: cephalic setae; b. ring: buccal ring; d. th: dorsal tooth; sv. th: ventrosublateral teeth; amph: amphidial fovea; n. ring: nerve ring; blb: bulb, Gub: gubernaculum; Spic: spicule.

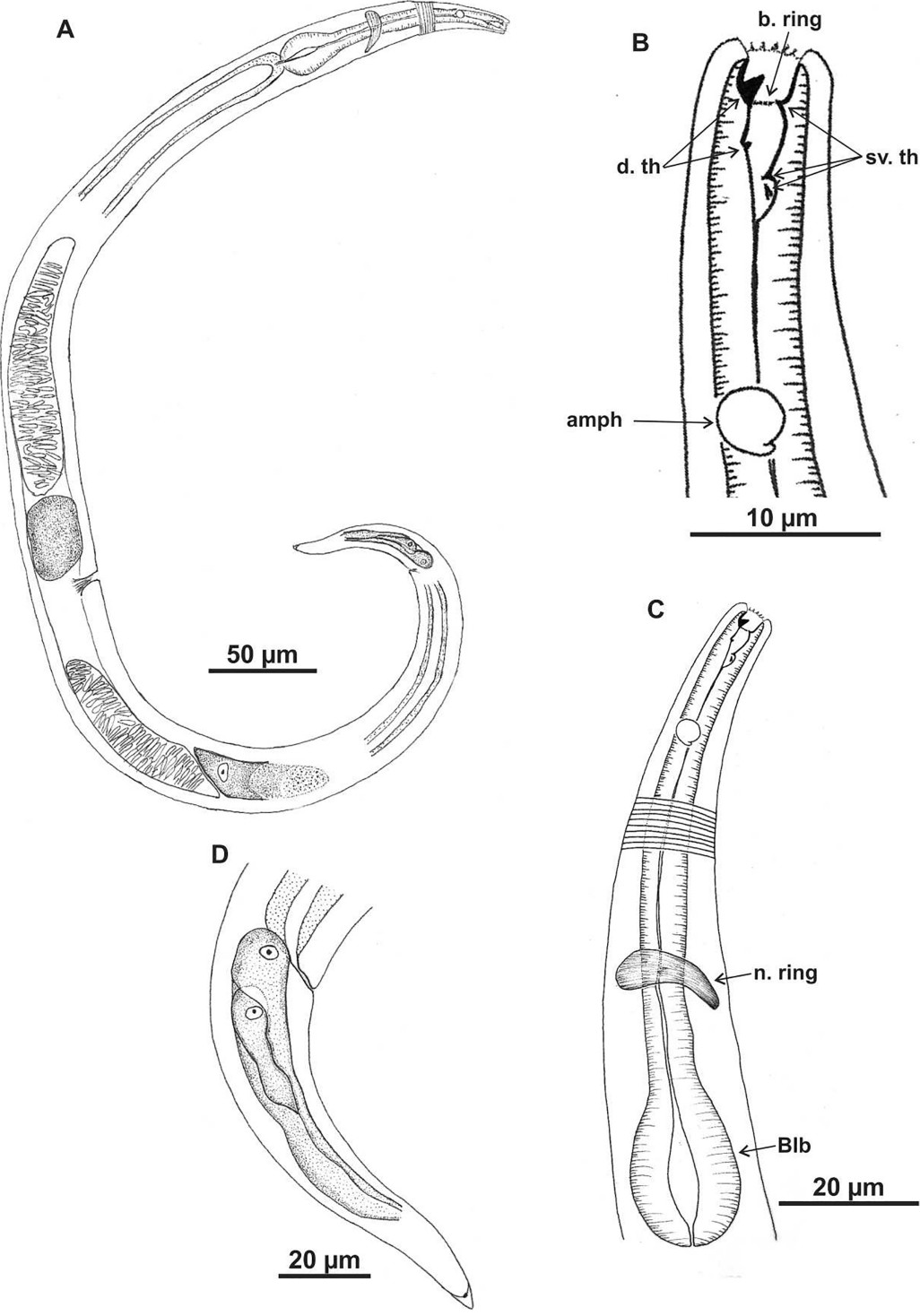

**Figure 5 *Microlaimus alexandri* sp. n.** Paratype female (MOUPE 0010): (A) habitus; (B) anterior end (buccal cavity, buccal ring and amphidial fovea); (C) anterior region (cuticle, nerve ring and bulb); (D) posterior region (tail). b. ring: buccal ring; d. th: dorsal tooth; sv. th: ventrosublateral teeth; amph: amphidial fovea; n. ring: nerve ring; blb: bulb.

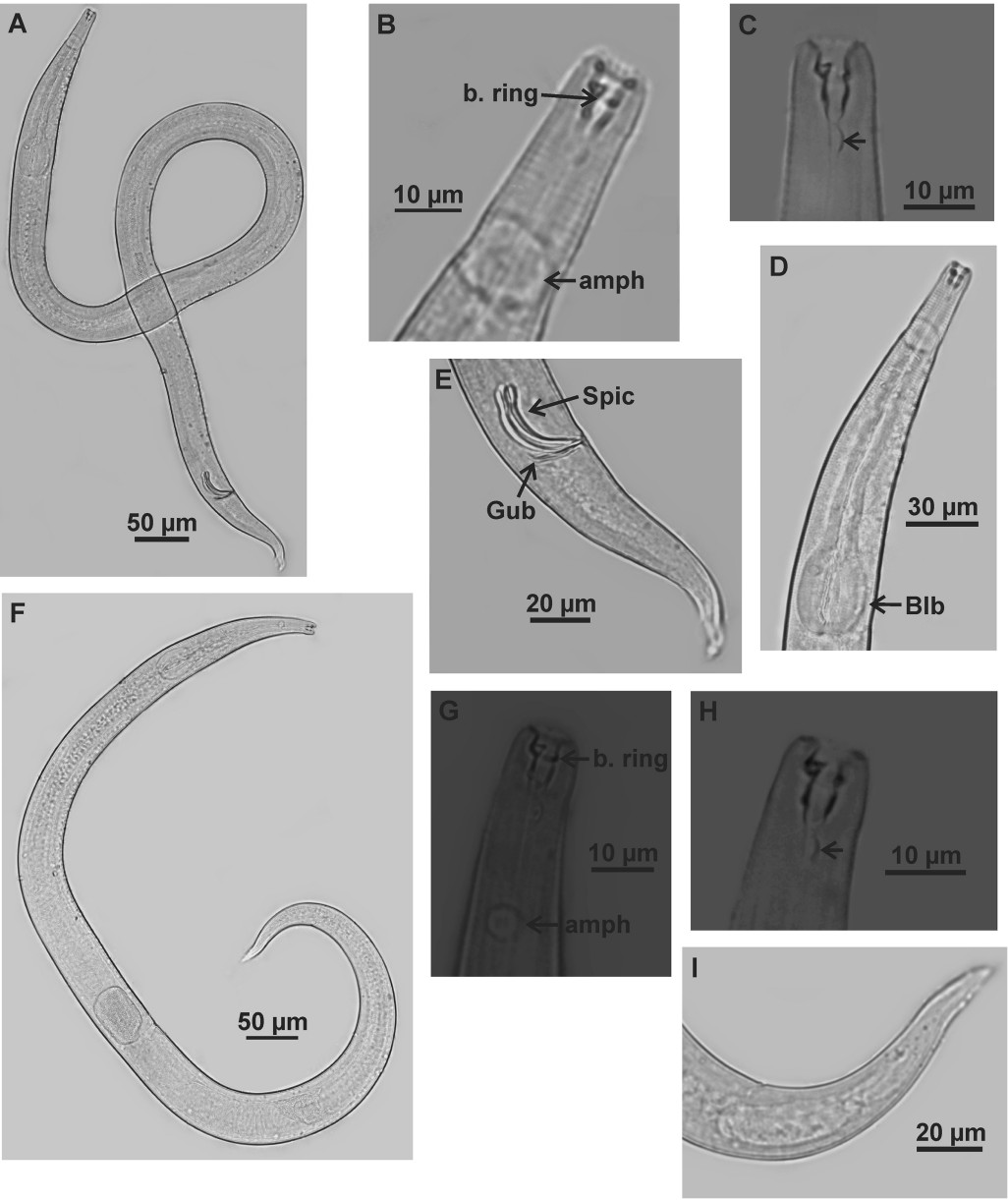

**Figure 6 *Microlaimus alexandri* sp. n.** Holotype (male): (A) habitus; (B) anterior end (buccal cavity, buccal ring and amphidial fovea); (C) posterior part of buccal cavity; (D) anterior region (bulb); (E) posterior region (spicules, gubernaculum and tail). Paratype female (MOUPE 0010): (F) habitus; (G) anterior end (buccal cavity, buccal ring and amphidial fovea); (H) posterior part of buccal cavity; (I) tail. b. ring: buccal ring; amph: amphidial fovea; blb: bulb; Gub: gubernaculum; Spic: spicule.

anteriorly positioned testes of different sizes; larger testis located to right of intestine and smaller testis to left. Elongated sperm (12–16 μm long and 3–5 μm wide). Spicules arched, with proximal portion cephalized, about 1× cloacal body diameter. Gubernaculum simple, lamellar, 0.4× spicule length. Tail conical, about 3× cloacal body diameter. Caudal glands present.

**Table 4 Measurements (in μm) of *Microlaimus alexandri* sp. n.**

|  | Holotype (Male) | Paratype Male | Paratype female |
|---|---|---|---|
| L | 1,146 | 1,173 | 1,089 |
| mbd | 34 | 28 | 48 |
| mbd/hd | 2.7 | 2.4 | 3.8 |
| a | 33.2 | 41 | 22.7 |
| b | 7 | 7.4 | 7.3 |
| c | 13.6 | 11.5 | 11.7 |
| amph. pos | 27 | 29 | 31 |
| amph | 16 | 22 | 5 |
| cbd/amph | 15.6 | 22.2 | 16.8 |
| Amph% | 100% | 100% | 32% |
| hd | 13 | 12 | 3 |
| els | NO | 1 | NO |
| cs | 7 | 6 | NO |
| cs/hd | 0.6 | 0.5 | NO |
| ph | 162 | 157 | 150 |
| blb | 28 | 24 | 26 |
| cbd/blb | 36 | 32.4 | 36 |
| blb% | 78% | 74% | 72% |
| n. ring | 109 | 94 | 94 |
| n. ring% | 67% | 59% | 63% |
| exc. p | NO | 107 | NO |
| exc. p% | NO | 68% | NO |
| abd | 27 | 28 | 24 |
| spic | 37 | 30 | NA |
| spic/abd | 1.4 | 1.4 | NA |
| gub | 18 | 15 | NA |
| v | NA | NA | 570 |
| V% | NA | NA | 52% |
| t | 84 | 102 | 93 |
| t/abd | 3.1 | 3.6 | 3.9 |

**Note:**
NA = not applicable, NO = not observed. For abbreviations see Table 1.

**Paratype female.** (Figs. 5 and 6; Table 4). Female generally similar to male. Body 1,089 μm long and maximum diameter 48 μm and wider than male (De Man ratio a = 22.7 *vs* 33.2–41 in the males). Anterior sensilla arrangement difficult to view. Amphidial fovea cryptocircular, 31 μm from anterior end (amph. pos/hd = 2.5). Diameter of amphidial fovea smaller than in male (5 μm *vs* 16-22 μm in males), occupying 32% of corresponding body diameter (100% in males), *i.e.*, sexually dimorphic. Buccal cavity, teeth and slightly cuticularized cuticular ring similar to those of male. Pharynx similar to that of male, with terminal bulb occupying 72% of corresponding diameter. Nerve ring located at 67% of pharynx length. Secretory-excretory and ventral glands not observed. Tail conical,

**Table 5 Comparison between *Microlaimus alexandri* sp. n. and morphologically similar species (only males).**

| Species | L | a | b | c | 2nd ceph ccl | Amph% | cs/hd | hd/mbd | spic/abd |
|---|---|---|---|---|---|---|---|---|---|
| *M. alexandri* **sp. n.** | 1,146–1,173 | 33.2–41 | 7–7.4 | 11.5–13.6 | papilliform | 100% male 32% female | 0.6 | 2.3–2.7 | 1.1 |
| *M. amphidius* | 720–852 | 32.1–39.4 | 7.9 | 9–11.3 | setiform | 100% male 30% female | 0.5 | 1.6 | 1.2 |
| *M. monstrosus* | 1,104–1,537 | 48–59 | 8.5–10.6 | 11.8–12.4 | papilliform | 100% male 45% female | 1 | 1.7 | 1.3 |
| *M. ostracion* | 1,150–1,310 | 42–45 | 7.7–8.7 | 9.9–11.5 | setiform | 33% male 33%female | 1.1 | 1.8–2.2 | 1.1 |

**Note:**
For abbreviations see Table 1.

3.8× anal body diameter. Caudal glands present. Reproductive system didelphic-amphidelphic, with outstretched ovaries located to right of intestine. Anterior and posterior genital branches measuring 170 and 177 μm, respectively. Vulva located at 52% of body length.

**Diagnosis.** *Microlaimus alexandri* **sp. n.** is characterized by the size of the amphidial fovea, occupying 100% of the corresponding area in males and 32% in females; the amphidial fovea is unispiral, in male, and cryptocircular, in female, and located more posteriorly from the anterior end, 2.1–2.4× cephalic diameter. The cephalic setae corresponding to 50–57% of the cephalic diameter. The buccal cavity is provided with five teeth, of which the anterior dorsal tooth is the most prominent, larger and heavily cuticularized than the other teeth; the second dorsal tooth, inserted in the median region of the buccal cavity, is very reduced in size. Of the ventrosublateral, teeth, one is situated at the level of the dorsal anterior tooth and two at the base of the oral cavity. The cuticular ring is slightly cuticularized and situated at the level of larger dorsal teeth base.

**Differential diagnosis.** (Table 5) The main diagnostic characteristic of *Microlaimus alexandri* **sp. n.** is the buccal cavity, which has five teeth, being a unique characteristic. *Microlaimus alexandri* **sp. n.** is similar to *M. amphidius* Kamran, Nasira & Shahina, 2009 in the size of the amphidial fovea (only in males), which occupies 100% of the corresponding body diameter. In addition, the new species shows similar values to *M. amphidius* in the De Man ratios a (33.2–41 *vs* 32.1–39.4 in *M. amphidius*) and b (7.0–7.4 *vs* 7.9). *M. alexandri* **sp. n.** differs from *M. amphidius* in having a longer body than *M. amphidius* (1,146–1,173 μm *vs* 720–852 μm); in the second circle of the cephalic sensilla arrangement (labial external circle), which is papilliform in the new species and setiform in *M. amphidius*; and the position of the amphidial fovea (more posterior) in *M. alexandri* **sp. n.** than in *M. amphidius* (amph. pos/hd = 2.1–2.4 *vs* 0.9 in *M. amphidius*). The new species resembles *M. monstrosus* Gerlach, 1953 in the size of the amphidial fovea (100% of the corresponding body diameter in males), body length (1,146–1,173 *vs* 1,104–1,537 μm in *M. monstrosus*), and De Man ratio c (11.5–13.6 *vs* 11.8–12.4 in *M. monstrosus*). However, *M. alexandri* **sp. n.** differs from *M. monstrosus* in the wider body (De Man ratio a 33–41 *vs* 48–59 in *M. monstrosus*), the smaller cephalic setae

(50–57% *vs* 100% of the corresponding cephalic diameter in *M. monstrosus*), and the position of the amphidial fovea (more posterior) in *M. alexandri* **sp. n.** than in *M. amphidius* (amph. pos/hd = 2.1–2.4 *vs* 0.9 in *M. monstrosus*). *M. alexandri* **sp. n.** is similar to *M. ostracion Schuurmans Stekhoven, 1935* in the presence of the cuticular ring in the buccal cavity and in the body length (1,146–1,173 *vs* 1,150–1,310 in *M. ostracion*). However, *M. alexandri* **sp. n.** has a striated cuticle without ornamentation whereas *M. ostracion* has cuticle ornamentation with longitudinal bars in striations, and other diferences such as the position of the amphidial fovea (more posterior) in *M. alexandri* **sp. n.** than in *M. ostracion* (amph. pos/hd = 2.1–2.4 *vs* 1.7 in *M. ostracion*), amphidial fovea size in relation to body diameter (100% *vs* 33% in *M. ostracion*), length of cephalic setae in relation to cephalic diameter (57% *vs* 110% in *M. ostracion*). Additionally, in *M. alexandri* **sp. n.** the anterior portion of the body is narrower than in *M. amphidius* and *M. monstrosus*; these three species have the maximum body diameter 2.7, 1.6 and 1.7 times greater than the cephalic diameter, respectively. Last, *M. alexandri* **sp. n.** differs from *M. amphidius*, *M. monstrosus* and *M. ostracion* in the number of teeth in the buccal cavity (five teeth *vs* three in *M. amphidius*, *M. monstrosus* and *M. ostracion*). The number and arrangement of the teeth in the buccal cavity is a unique characteristic of *M. alexandri* **sp. n.**

*Microlaimus vitorius* **sp. n.**
(Figs. 7–9; Tables 6 and 7)

**Type material.** Holotype male (MOUFPE 0011), paratype female (MOUFPE 0012), 4 male paratype (438–441 NM LMZOO-UFPE) 1 female paratype (442 NM LMZOO-UFPE).

**Type locality.** Campos Basin, Rio de Janeiro, Brazil. Holotype male and paratype female 21°18'40"S 40°47'46"W (25 m depth). July 2009.

**Etymology.** The specific epithet *vitorius* is given in honor of the late Professor Verônica Gomes da Fonsêca-Genevois. Veronica is the latinized form of the name Berenice, which in Macedonian means bearer of victory (Greek: phere-nikē).

**Description. Holotype male** (Figs. 7 and 9; Table 6). Body 1696 μm long. Maximum body diameter 59 μm, corresponding to 3.7× cephalic diameter at level of cephalic setae. Cuticle striated from insertion of cephalic setae. Four lateral rows of hypodermal glands associated with small pores that extend longitudinally from pharynx to beginning of tail. Anterior sensilla arrangement in general pattern of genus, in three distinct cycles; inner labial and external labial papilliform, cephalic setae measuring 52% of corresponding body diameter. Amphidial fovea cryptocircular, located in posterior portion of buccal cavity, occupying 52% of corresponding body diameter. Buccal cavity cuticularized, with folds in its first portion; provided with three cuticularized teeth, one dorsal and two ventrosublateral, about same size. Pharynx involving buccal cavity and terminating in oval bulb. Nerve ring not observed. Secretory-excretory pore, distance from anterior end equivalent to 41.5% of pharynx length. Ventral gland not observed. Male reproductive

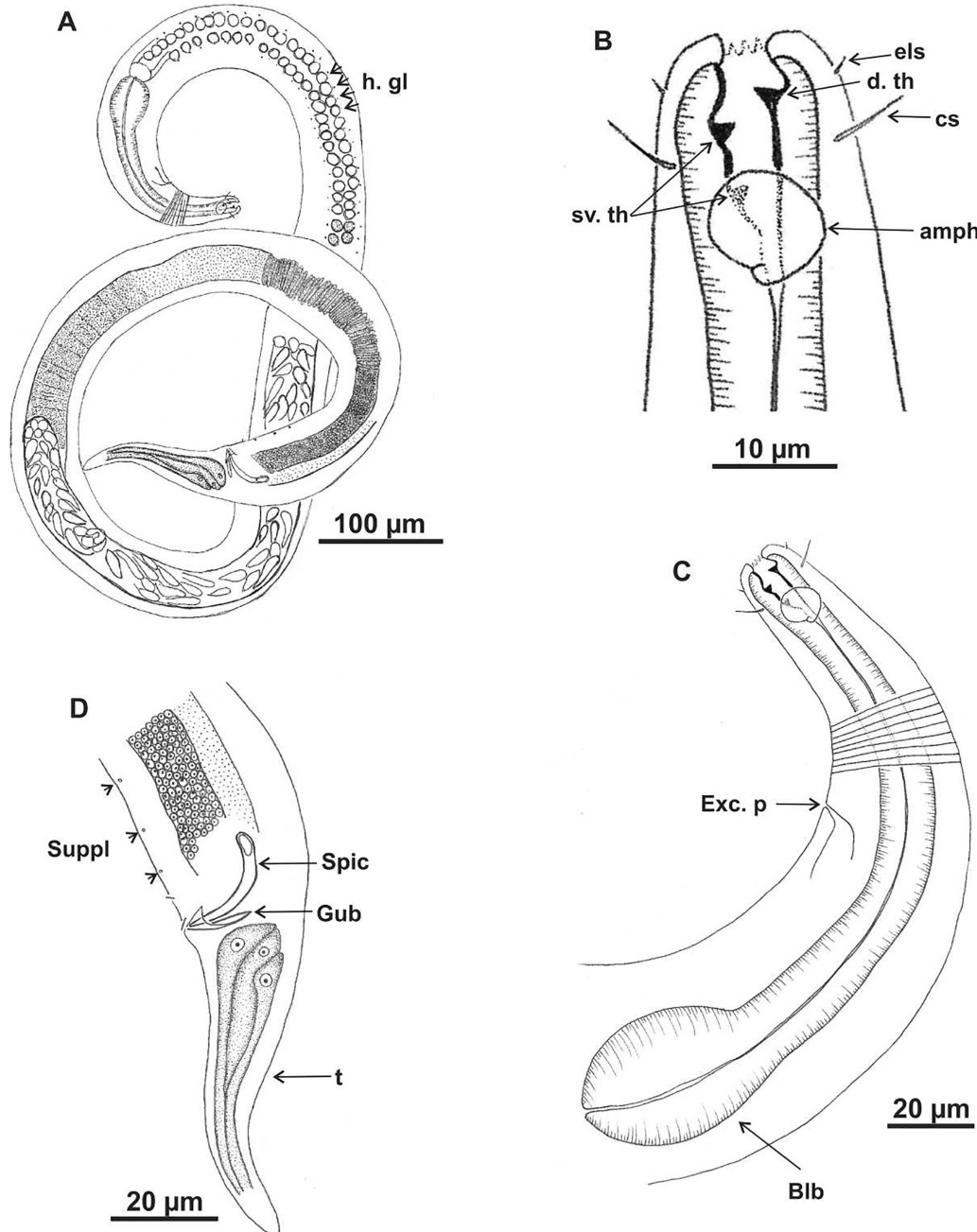

**Figure 7 *Microlaimus vitorius* sp. n.** Holotype (male): (A) habitus (hypodermal glads); (B) anterior end (buccal cavity and amphidial fovea); (C) anterior region (cuticle, secretory-excretory pore and bulb); (D) posterior region (tail, spicule, gubernaculum, and pre-cloacal seta and pores). h. gl: hypodermal glads; els: external labial setae; cs: cephalic setae; d. th: dorsal tooth; sv. th: ventrosublateral teeth; amph: amphidial fovea; Exc. p: secretory-excretory pore; blb: bulb; Gub: gubernaculum; Spic: spicule; Suppl: pre-cloacal supplements.               

system with two testes extending in opposite directions, larger anterior testis to right of intestine and smaller posterior testis to left of intestine. Sperm fusiform (16–33 μm long and 6–8 μm wide). Spicules arched, 1× cloacal body diameter. Gubernaculum lamellar,

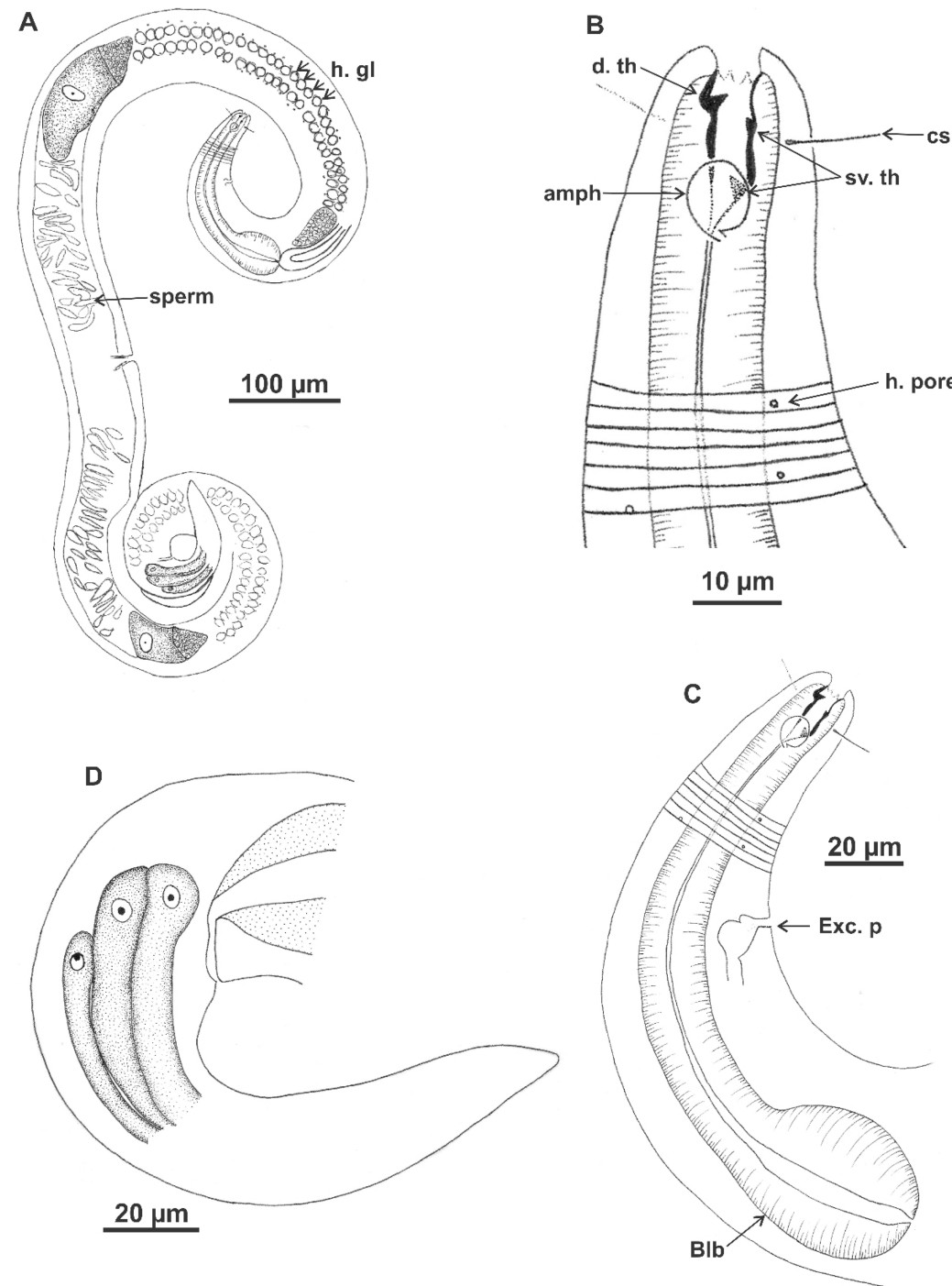

**Figure 8 *Microlaimus vitorius* sp. n.** Paratype female (MOUPE 0012): (A) habitus (hypodermal glands); (B) anterior end (buccal cavity, amphidial fovea and hypodermal pore); (C) anterior region (cuticle, secretory-excretory pore and bulb); (D) tail. h. gl: hypodermal glads; esl: external labial setae; cs: cephalic setae; d. th: dorsal tooth; sv. th: ventrosublateral teeth; amph: amphidial fovea; h. pore: hypodermal pore; Exc. p: secretory-excretory pore; blb: bulb; sperm: spermatozoids.

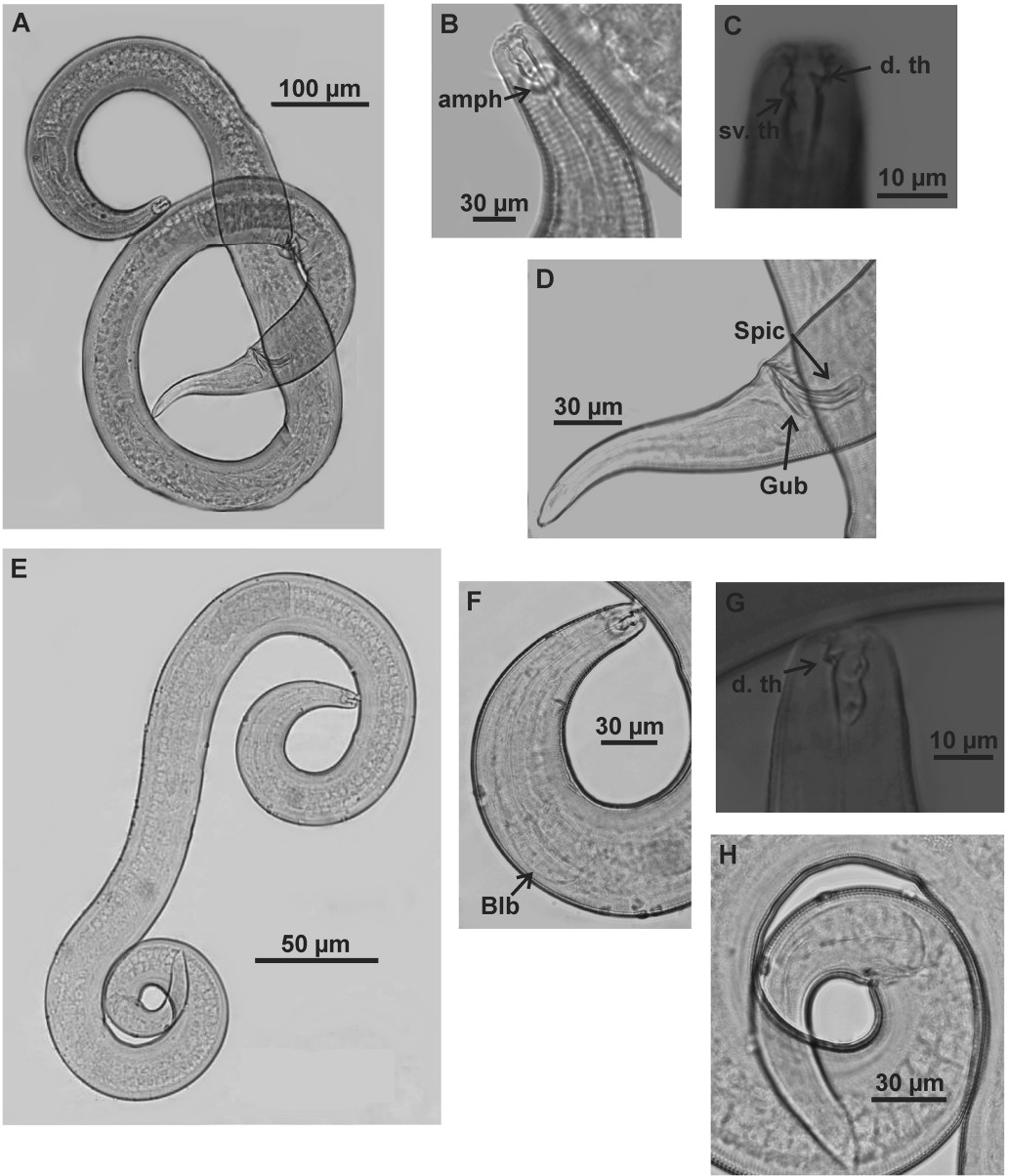

**Figure 9** *Microlaimus vitorius* **sp. n.** Holotype (male): (A) habitus (hypodermal glands); (B) anterior region (buccal cavity and amphidial fovea); (C) buccal cavity (dorsal and ventrosublateral teeth); (D) posterior region (spicules, gubernaculum and tail). Paratype female (MOUPE 0012): (E) habitus (hypodermal glands); (F) anterior region (bulb); (G) buccal cavity; (H) posterior region (hypodermal glands and tail). d. th: dorsal tooth; sv. th: ventrosublateral teeth; amph: amphidial fovea; Gub: gubernaculum; Spic: spicule.

with triangular base. One pre-cloacal seta and three small pre-cloacal pores. Tail conical (134 μm) with three glands.

**Paratypes females** (Figs. 8 and 9; Table 6). Female similar to male. Body 1,590–1,745 μm long and maximum diameter 70–86 μm. Four lateral longitudinal rows of hypodermal glands associated with small pores, in same arrangement as in male. External labial papillae

**Table 6 Measurements (in μm) of *Microlaimus vitorius* sp. n.**

| | Holotype (Male) | Paratype Male 1 | Paratype Male 2 | Paratype Male 3 | Paratype Male 4 | Paratype Female (MOUFPE 0012) | Paratype Female 1 | Paratype Female 2 |
|---|---|---|---|---|---|---|---|---|
| L | 1,696 | 1,625 | 1,922 | 1,726 | 1,751 | 1,745 | 1,683 | 1,590 |
| mbd | 59 | 54 | 61 | 69 | 67 | 70 | 78 | 86 |
| mbd/hd | 3.7 | 3 | 3.5 | 3.4 | 3.5 | 3.3 | 4.2 | 4.8 |
| a | 28.6 | 30.1 | 31.2 | 25 | 26 | 25 | 21.5 | 18.4 |
| b | 8.8 | 8.7 | 9.8 | 8.4 | 9.4 | 9.5 | 8.1 | 7.6 |
| c | 12.6 | 12 | 13.9 | 12.5 | 12 | 13.5 | 10.5 | 9.5 |
| amph. Pos | 12 | 9 | 14 | 16 | 12 | 13 | 14 | 13 |
| amph | 10 | 11 | 9 | 9 | 11 | 7 | 9 | 9 |
| cbd/amph | 19.2 | 21.6 | 21 | 24 | 23.4 | 22.2 | 21.7 | 20.4 |
| Amph% | 52% | 53% | 43% | 37.5% | 49% | 32% | 40% | 44% |
| hd | 16 | 18 | 17 | 20 | 19 | 21 | 19 | 18 |
| els | 1 | 1 | NO | NO | 1 | NO | NO | NO |
| cs | 8 | 10 | 13 | 14 | 13 | 12 | 10 | 9 |
| cs/hd | 0.5 | 0.5 | 0.7 | 0.7 | 0.7 | 0.6 | 0.6 | 0.5 |
| ph | 194 | 186 | 195 | 203 | 186 | 184 | 208 | 210 |
| blb | 34 | 31 | 37 | 32 | 35 | 36 | 36 | 40 |
| cbd/blb | 51.6 | 48.6 | 52.8 | 60 | 63.6 | 54.6 | 57 | 59.2 |
| blb% | 66% | 64% | 69% | 68% | 55% | 66% | 63% | 68% |
| n. ring | NO | 106 | NO | NO | NO | NO | NO | NO |
| n. ring% | NO | 56.9% | NO | NO | NO | NO | NO | NO |
| exc. p | 80 | 74 | 83 | 83 | 80 | 74 | NO | 83 |
| exc. p% | 41.5% | 40% | 43% | 41% | 43% | 40% | NO | 40% |
| abd | 50 | 43 | 57 | 54 | 57 | 39 | 32 | 43 |
| spic | 50 | 45 | 55 | 46 | 54 | NA | NA | NA |
| spic/abd | 1 | 1 | 0.9 | 0.9 | 0.9 | NA | NA | NA |
| gub | 27 | 21 | 22 | 21 | 20 | NA | NA | NA |
| v | NA | NA | NA | NA | NA | 920 | 920 | 840 |
| V% | NA | NA | NA | NA | NA | 53% | 53% | 53% |
| t | 134 | 126 | 138 | 138 | 146 | 129 | 160 | 168 |
| t/abd | 2.7 | 2.9 | 2.4 | 2.5 | 2.6 | 3.3 | 5 | 3.9 |

Note:
NA = not applicable, NO = not observed. For abbreviations see Table 1.

**Table 7 Comparison of *Microlaimus vitorius* sp. n. and morphologically similar species (only males).**

| Species | L | a | b | c | 2nd ceph ccl | Amph% | amph. pos/hd | cs/hd | hd/mbd | spic/abd |
|---|---|---|---|---|---|---|---|---|---|---|
| *M. vitorius* **sp. n.** | 1,696–1,921 | 28.6 | 8.8 | 12.6 | papilliform | 37.5–53 | 0.4–0.7 | 0.8 | 3.7 | 1 |
| *M. acinaces* | 945–1,227 | 29–38.5 | 6.5 | 11.9–15 | setiform | 58–65 | 0.8–1 | 0.5 | 2.3 | 1–1.3 |
| *M. cyatholaimoides* | 700–1,000 | 22–31 | 6.8–7.8 | 9.7–12.7 | papilliform | 35 | 1.5–1.7 | 0.4 | 2.9–3.6 | 1.5 |
| *M. discolensis* | 425–560 | 15.2–18.7 | 6.6 | 8.5–10.6 | setiform | 59 | 1.1 | 0.5 | 1.9–2.1 | 1.3 |
| *M. porosus* | 380–644 | 21 | 4.9 | 5.4 | papilliform | 40–50 | 1.9 | 0.4 | 2.2 | 1.8 |
| *M. parviporosus* | 360–415 | 30.2–25.9 | 4.6–5.2 | 7.3–8.3 | setiform | 55–67 | 1.6 | 0.2 | 1.8–2 | 1.5 |

Note:
For abbreviations see Table 1.

difficult to view. Cephalic setae corresponding to 50–57% of cephalic diameter. Amphidial fovea cryptocircular, smaller than in male, occupying 32–44% of corresponding body diameter, located in similar position to male. Buccal cavity and teeth also similar to those of male. Pharynx similar to that of male, with terminal bulb occupying 63–68% of corresponding diameter. Nerve ring not observed. The secretory-excretory pore in similar position to male, 7–83 μm from anterior end and equivalent to 40% of pharynx length. Ventral gland located immediately posterior to pharynx. Tail with same shape and measurements as in male. Three caudal glands. Reproductive system didelphic-amphidelphic, outstretched ovaries located to right of intestine. Anterior and posterior genital branches measuring 280–320 and 265–324 μm, respectively. Sperm present in uterus. Vulva located 840–920 μm from anterior end, corresponding to 53% of body length.

**Diagnosis.** *Microlaimus vitorius* **sp. n.** has four lateral longitudinal rows of hypodermal glands that open through small pores and extend from the pharynx to the beginning of the tail in both sexes. Cephalic setae comprise 50–72% of corresponding body diameter. Amphidial fovea in posterior portion of buccal cavity, accounting for 37.5–52% of corresponding body diameter in male and 32–44% in female. Buccal with three cuticularized teeth, one dorsal and two ventrosublateral, about same size. Male with one pre-cloacal seta and three small pre-cloacal pores. Gubernaculum lamellar, with triangular base.

**Differential diagnosis** (Table 7). Males of *Microlaimus vitorius* **sp. n.** and *M. acinaces* *Warwick & Platt, 1973* are similar in the large, heavily cuticularized buccal cavity and well-developed dorsal tooth, and the presence of a pre-cloacal seta and small pre-cloacal supplements. *Microlaimus vitorius* **sp. n.** differs from *M. acinaces* in cephalic sensilla arrangement (external labial circle is papilliform in the new species *vs* setiform in *M. acinaces*), longer cephalic setae (79% *vs* 50% of the corresponding cephalic diameter in *M. acinaces*) and the presence of four lateral longitudinal rows of glands associated with small pores in *M. vitorius* **sp. n.** Additionally, *M. acinaces* has four longitudinal rows of somatic setae, two ventrosublateral, rows of pre-cloacal ducts associated with subcuticular glands, and the tail with ventrosublateral, rows of setae, characteristics not observed in *M. vitorius* **sp. n.** *Microlaimus vitorius* **sp. n.** and *M. cyatholaimoides* are similar in the pore-like pre-cloacal supplements and the four longitudinal rows of lateral glands. However, in this latter species, the glands are associated with setae. These species also differ in the amphidial fovea, which is located more anteriorly in the new species (amph. pos/hd = 0.4–0.7 in *M. vitorius* **sp. n.** *vs* 1.5–1.7 in *M. cyatholaimoides*) and the length of the cephalic setae (79% *vs* 40% of the cephalic diameter in *M. monstrosus*). Another important difference between the two species is the presence of testes positioned in opposite directions in *Microlaimus vitorius* **sp. n.** *vs* two anterior testes in *M. cyatholaimoides*. *Hopper & Meyers (1967)* and *Muthumbi & Vincx (1999)* considered the rows of hypodermal glands along the body as a species-level diagnostic character. *Microlaimus vitorius* **sp. n.** shares this character with *M. cyatholaimoides* and three other

species: *M. discolensis Bussau, 1993*, *M. porus Bussau, 1993* and *M. parviporosus Miljutin & Miljutina, 2009*. However, *M. vitorius* **sp. n.** differs from these last three species in the longer body (1696–1921 in *M. vitorius* **sp. n.** *vs* 425–560 μm in *M. discolensis*, 380–644 μm in *M. porus* and 360–415 μm in *M. parviporosus*), longer cephalic setae (79% of the cephalic diameter in *M. vitorius* **sp. n.** *vs* 55% in *M. discolensis*, 40% in *M. porus* and 20% in *M. parviporosus*), proportion of spicules in relation to cloacal body diameter (1.0× in *M. vitorius* **sp. n.** *vs* 1.3 in *M. discolensis*, 1.8 in *M. porus* and 1.5 in *M. parviporosus*) and the position of the amphidial fovea (amph. pos/hd = 0.4–0.7 in *M. vitorius* **sp. n.** *vs* 1.1 in *M. discolensis*, 1.9 in *M. porus* and 1.6 in *M. parviporosus*).

## DISCUSSION

The shape of the setae (papilliform or setiform) of the second and third circles of the cephalic arrangement, the relationship between the length of the cephalic setae and the cephalic diameter, the diameter of the amphidial fovea in the corresponding region of the body (%) and its position in relation to the anterior extremity of the body provided important taxonomic information that was used to distinguish *Microlaimus* species. The percentage occupied by the amphidial fovea was previously indicated as an interspecific morphological variation for *Microlaimus* by *Armenteros, Vincx & Decraemer (2010)*. Furthermore, *Armenteros et al. (2009)* argued that morphometry could be used in studies of phylogenetic relationships. These authors also suggested the possibility of evaluating the relationship between morphological plasticity and ecological success in free-living marine nematodes, based on a model proposed by *Hollander (2008)*.

Interspecific differences in the species of the genus *Microlaimus* were observed for the number of testes, their position in the body (anterior or posterior) and their shape (outstretched or reflexed). Most species have two testes extending in opposite directions, and of the same or different sizes (the posterior testis may be shorter). Two anterior testes may be present, as in *M. cyatholaimoides De Man, 1922*, as noted by *Pastor de Ward (1989)*, and in *M. campiensis* **sp. n.** and *M. alexandri* **sp. n.** Only six species have one testis, which is anterior in *M. martinezi* (*Miljutin & Miljutina, 2009*), *M. nanus Blome, 1982*, *M. nympha* (*Bussau, 1993*), *M. texianus Chitwood, 1951* and *M. westindicus* (*Kovalyev & Miljutina, 2009*), and posterior and reflexed in *M. capillaris Gerlach, 1957*, as noted by *Jensen (1989)* for this last species. No information on these structures was provided in the species descriptions for approximately 30 species of *Microlaimus. Tchesunov (2014)* stated that this is common in the literature on Microlaimidae. However, despite this lack of information, we included the interspecific variability of this structure in the diagnosis of the genus.

Finally, it should be noted that the species *M. alexandri* **sp. n.**, *M. campiensis* **sp. n.**, and *M. vitorius* **sp. n.** were recorded only on the continental shelf (depth 25–50 m) of the Campos Basin, during a taxonomic study that also sampled on the continental slope (up to 3,000 m) of the basin (*Esteves et al., 2017*; *Fonsêca-Genevois et al., 2017*). This observation may indicate a possible bathymetric restriction of these species.

## CONCLUSION

Based on the high richness of *Microlaimus* species, it is surprising to see how much of this diversity may still be unknown. Three new species of this genus were found in the continental shelf region off the Brazilian coast. Therefore, this result reinforces the importance of taxonomic studies of nematodes, one of the most abundant and diverse metazoan group.

## ACKNOWLEDGEMENTS

Sincere thanks are also due to Dr. Rafael Bendayan de Moura and MSc. Alex Manoel for help in obtaining the images and to Dr. Janet W. Reid, JWR Associates, for the English revision.

### Funding

This work was supported by the 'Habitats Project—Campos Basin Environmental Heterogeneity' of CENPES/PETROBRAS by collecting the samples and providing the opportunity to study them. Andre M. Esteves (CNPq 310249/2019-8) was supported by a research fellowship from the Conselho Nacional de Desenvolvimento Científico e Tecnológico (CNPq). The funders had no role in study design, data collection and analysis, decision to publish, or preparation of the manuscript.

### Grant Disclosures

The following grant information was disclosed by the authors:
CENPES/PETROBRAS.
Conselho Nacional de Desenvolvimento Científico e Tecnológico (CNPq): CNPq 310249/2019-8.

### Competing Interests

The authors declare that they have no competing interests.

### Author Contributions

- Rita C. Lima conceived and designed the experiments, performed the experiments, analyzed the data, prepared figures and/or tables, authored or reviewed drafts of the paper, and approved the final draft.
- Patricia F. Neres conceived and designed the experiments, performed the experiments, analyzed the data, prepared figures and/or tables, authored or reviewed drafts of the paper, and approved the final draft.
- Andre M. Esteves conceived and designed the experiments, prepared figures and/or tables, authored or reviewed drafts of the paper, and approved the final draft.

### Data Availability

    All data used for the descriptions is available in the tables.

## New Species Registration

The following information was supplied regarding the registration of a newly described species:

Publication LSID: urn:lsid:zoobank.org:pub:B4106DEE-2BC2-48D2-9BE5-41573F76FB25

*Microlaimus alexandri* urn:lsid:zoobank.org:act:265202E4-A128-41E9-B83A-9EFCE7B773BD

*Microlaimus campiensis* urn:lsid:zoobank.org:act:E8644075-41C6-433A-A46E-0643F9CD5EBD

*Microlaimus vitorius* urn:lsid:zoobank.org:act:7564370C-6CC7-41BB-BA3B-769321935E9D

## Supplemental Information

Supplemental information for this article can be found online at http://dx.doi.org/10.7717/peerj.12734#supplemental-information.

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
