# Peer review of "Three new species of Microlaimus (Nematoda: Microlaimidae) from the South Atlantic"

_PeerJ, doi:10.7717/peerj.12734_

## Round 0.1 · original submission · Major Revisions

I have heard back from three reviewers, each of whom sees merit in your work while offering numerous constructive comments than can help you make your paper better. Please take a look at their comments; I look forward to seeing a revised version of your work.

·

Basic reporting

English is generally correct and not ambiguous.
New features and characteristics are well and detailed described, but it is not always possible to check them because raw data are not submitted and some photos are blurry. To get high quality photos of meiofaunal animals is not always easy but it is an important goal in research diversity studies.
Drawings are clear and well done.
Symbol legends can be added to all the photos/drawings in order to help the reader finding the described features.
The submission is self-contained and the results truly clear.
Adding a Conclusions section can help the reader to understand why your research should matter.

20 An amendment to
21 the diagnosis of the genus is proposed.
Can you explain why you propose an amendment of the genus?

35 it was the most abundant genus, in terms of the number of individuals, of the family
How many individuals were found?

58 Although different in surface area, the two samplers have a similar design that aims
59 at maintaining the integrity of the vertical sediment structure
The meaning is unclear, can you explain this part?

A general background/context is provided.

26 (1922). This group is predominantly marine, although its type species M. globiceps De Man,
27 1880 is one of the few freshwater species.
M. globiceps is a terrestrial, marine and freshwater species, as many others (Nemys - https://nemys.ugent.be/aphia.php?p=taxdetails&id=121080 )

63 Each sediment sample was transferred to a plastic flask and fixed in 10% formalin
64 buffered with borax.
Adding a reference for this part can be useful in understanding the methodology.

349 sperm 350 present in uterus. How can you determine the presence of the sperm?

Experimental design

This research is in line within Aims and Scope of the journal. Questions are well defined and significant and results contribute in filling a knowledge gap in nematodes taxonomy.
The research is clear but, in some parts, experimental methods can be described more in detail providing references that can help the reader and potential replicators.

58 Although different in surface area, the two samplers have a similar design that aims
59 at maintaining the integrity of the vertical sediment structure…
and
62 three replicated sediment quadrats (10 x 10 cm) for meiofauna (0–10 cm depth) were
63 sampled.
Samplers or quadrats or both methods were used? How? To add references and explain how the quadrats were used can help in understanding the methodology.

65 Meiofauna samples were sieved through a 500-μm mesh…
A 500-micrometers mesh sieve is inadequate considering that nematodes are commonly bigger. A 1 mm or 2 mm mesh sieve is usually used instead as in ‘Influence of mesh size and core penetration on estimates of deep-sea nematode abundance, biomass, and diversity’ D.Leduc, P.K.Probert, S.D.Nodder
Can you explain why you choose to use this kind of sieve?

Validity of the findings

There are some features not identifiable in the photos, as for the cephalic setae structure.
In all images and photos adding abbreviations can be useful to speed up reading.
This is a suggestion: I think it is better to include all the features mentioned in the paper in the relative table for the same reason as before.

TAB.1
t or T and v or V (check in all tabs)

TAB.2
Use sign = after NA and NO instead of commas as in TAB.4 and TAB.6
Check significant figures (in all tabs)
Add cs/hd

TAB.3
Add space between ‘and’ and ‘morphologically’

TAB. 4
Add mbd/hd, hd/cs, t/abd because are mentioned in the description (same considerations for female allotype)
cs(=7,2) is not 50% of hd(=12,6)

TAB.5
For M. monstrosus write the length range 1104−1537 μm instead of 1240 as for the other examples

TAB.6
Add mbd/hd, cbd/cs (male), cs/hd (female)

TAB.7
Add amph. pos/hd = 0.4–0.7 in M. vitorius vs 1.5–1.7x in M. cyatholaimoides, 1.1 in M. discolensis, 1.9 in M. porosus and 1.6 in M. parviporosus

FIG.3
137 labial papillae and four cephalic setae, in different cycles. Cephalic setae corresponding
179 cuticularized, and with large teeth. However, M. campiensis sp. n. can be differentiated from 180 M. affinis by the cephalic setae length (46–50% of head diameter vs 37–43% in M. affinis).
Cephalic setae are not visible in the photos, thus it can be difficult for a reader to understand the description.
146 Seven pre-cloacal 147 papilliform supplements
Only 6 supplements are highlighted in the photo, but more than 6 are visible as described.
165 Presence of two glands, one on each ovary branch.
As for cs, glands may be difficult to check.

FIG.6
FIG.6 C cs visible but not clear
FIG.6 C and F show the same feature
FIG.6 G tail not visible

FIG.9
Cephalic setae not visible

Additional comments

Data and discussion do not contradict each other and there is sufficient data to support the conclusion. It can be improved by adding more references and clearer photos (with symbol legends, for male holotypes and female allotypes). Finally, fixing minor mistakes in the tables can be helpful in refining the work.

Reviewer 2 ·

Basic reporting

General comments:
I congratulate the authors for making a valuable contribution to our knowledge of a complex genus, Microlaimus. My main reservation about the manuscript, which must be addressed by the authors, is that there must be more detailed information on the status of the genus, the known diversity (eg, how many species are there in the genus?), recent revisions and descriptions, and lists of species. The authors need to consult Kovalyev and Tchesunov (2005) and Leduc (2016), as well as the other authors they cite at the end of their Introduction (not just cite them but give a summary of their key findings, and also include an updated species list). At present, the authors do not convince me that they have made exhaustive comparisons with all valid species of the genus, because they do not demonstrate a familiarity with all of the relevant literature. Microlaimus is a large genus and one needs to be diligent in checking all of the literature to avoid describing species which are in fact already named. The authors should also realise that the Microlaimoidea are now classified within their own order, the Microlaimida (see Leduc et al. 2018 ZJLS), and not the Desmodorida.
I wish someone would review the genus and divide it into groups that would facilitate identification, and maybe devise a key. However I do not ask that the authors necessarily do this.

Specific comments:
Line 105: nematodes don’t have a head, they have a cephalic region.
Line 109: The appropriate taxonomic authority for Microlaimus dicolensis has been established to be Bussau, 1993, not Bussau & Vopel, 1999. See Holovachov (2020) Bionomina.
Line 110: a unispiral amphid has a circular shape by definition
Line 113: replace “sclerotized” by “cuticularized” here and throughout manuscript. Clarify whether this ring surrounds the whole buccal cavity (what do you mean by “at base of dorsal tooth?)
Line 116: males do not have an anus, they have a cloaca. Correct here and throughout manuscript.
Lines136-138: are the external labial sensilla papilla or setae? Choose one do not use both
139: posterior to, not “after”. Correct here and throughout manuscript
140: “with 12 folds at upper end” - do you mean “Cheilostoma with twelve longitudinal folds”?
Line 141: The correct term is ventrosublateral, not sublateral, see Coomans 1979 - A PROPOSAL FOR A MORE PRECISE TERMINOLOGY OF THE BODY REGIONS IN THE NEMATODE
Line 142: round number to 78% Cannot possibly be that precise. Same for line 139, 157, 161, 164 etc…
Line 153: You write “generally similar”, but is this a description of a single female? I don’t think that allotypes are used anymore, you need to check the Code.
Line 165: these don’t look like glands to me, more like sphincter muscles. Why would glands surround each genital branch? Their function cannot be to push the egg along!
Line 172: “enlarged proximal end” – do you mean a capitulum?
Line 173: delete “strangler”, and see my comment above about muscle vs gland
Line 181: only one spicule? Usually they have two.
Line 244: what do you mean by “situated in same row”? clarify.
Line 248: provide dimensions of sperm for all spcies
Line 248: “with proximal portion cephalised” – you mean capitulum present?
You need to provide a better and larger drawing of the buccal cavity showing the unusual structure in M. alexandri (5 teeth)
Line 272-273: “The cuticular ring…” what do you mean by “same row” the ring needs to be better shown in drawing. Too small. You also need to show the longitudinal slits (openings of epidermal glands) in the drawings.
Line 401: testes, not testicles
Line 412: should be: “However, despite this lack of information…”
Table 2: no decimal points on the percentages. Here and throughout manuscript
Figure 1: the vas deferens near the spicules looks a bit like a posterior testis? On Figure 1A

Experimental design

see above

Validity of the findings

see above

Reviewer 3 ·

Basic reporting

Three new species of Microlaimus (Nematoda:
Microlaimidae) from the South Atlantic (#53043)
Review Report

Thank you for the opportunity to review the manuscript on description of three Microlaimus species as entitled above.
This excellent work on taxonomy of marine nematode, it adds knowledge for biodiversity and improves the list of known species. The manuscript is well written and especially the line drawings are well done and clear. The tables are done with a lot of details that helps the reader compare the different species described here and with others. The descriptions of each species are elaborated and the diagnosis and differential diagnosis clearly set out.
I have however a few comments and 1 or 2 suggestions
1. Line 33-34 there is indication of Microlaimus occurring along the continental shelf and slope, yet all the three species described were sampled from 25m. And so as far as the species described here are concerned, there is evidence of occurrence at continental shelf.
2. Line 40-43: The statement suggests that Molgolaimus belongs to Microlaimidae why it belongs to Desmodoridae and more recent work by Leduc et al (2019) suggest puts it in Chromadorida.
3. One weakness on the descriptions that runs throughout the three species is the focus of description and measurements on the male holotype alone and the female allotype. It would be best to consider all the individuals and thus report measurements as a range rather than single figure but keeping the male and female descriptions separate as they are currently is okay.
4. Counter-check table referencing in the text. For instance, Line 134: Microlaimus campiensis should be Table 2 not 3 and this error runs through out the text.
5. Line 135: It is easier to read the text and follow when actual measurements are mentioned, where necessary, before the relative sizes. For instance, ‘Maximum body diameter corresponding to 1.8x head diameter’, neither mbd, nor head diameter is given. I acknowledge all information is in the tables but some guiding data is necessary in the text.
6. The structuring of sentences needs to be counter-checked. For instance, the sentence on 142-143 ‘ Nerve ring located at level of secretory-excretory pore, distance from anterior end equivalent to 60% of pharynx length’. Line 164: ‘…… intestine, comprising 50% of body length’ sentence not clear.
7. Line 174: I could not see the comb-like glands in the female ovary. It would be nice to show them either in the plate or line drawing more clearly. The description of the glands as ‘strangler’ needs to be elaborated.
8. Line 221: Microlaimus alexandri sp. n. I have a problem with the number of individuals available for description of this species. With only one female and sexual dimorphism in the amphid size and location, one is left wondering whether the three individuals are really same species. The author needs to decide how to deal with that challenge of the numbers-delay publishing until enough material is found or describe only males and assume females were not found.
Thank you again for the opportunity

Agnes Muthumbi, University of Nairobi

Experimental design

Not applicable

Validity of the findings

Very useful results in the light of the need to document biodiversity in ecosystems

---

## Round 0.2 · Minor Revisions

I have heard back again from two of the previous reviewers, who find your work much improved with only minor comments to help you further improve your work. I imagine you can complete these final revisions quickly, and look forward to seeing your new version.

·

Basic reporting

English is fluent and professional
Clearer figures were added
Highlighted mistakes were fixed
Symbols were added to the respective figures
A conclusion chapter was added, but there are some typos:
• 429. Microlaimus in italics
• 429. species
• 423-427. Conclusion before acknowledgements

Experimental design

Now it is more detailed

Validity of the findings

Figures and table were corrected

Reviewer 2 ·

Basic reporting

see attachment

Experimental design

see attachment

Validity of the findings

see attachment

Annotated reviews are not available for download in order to protect the identity of reviewers who chose to remain anonymous.

---

## Round 0.3 · accepted · Accept

Thank you for your final revisions. I am happy to move this work into production, and look forward to seeing the published version of your work.